# Acute psychological impact of coronavirus disease 2019 outbreak among psychiatric professionals in China: a multicentre, cross-sectional, web-based study

Xin Guo [ORCID],[1] Robert McCutcheon,[2] Toby Pillinger,[2] Atheeshaan Arumuham,[2] Jianhua Chen,[3] Simeng Ma,[1] Jun Yang,[4] Ying Wang,[1] Shaohua Hu,[5] Gaohua Wang,[1] Zhong-Chun Liu [ORCID] [1]

For numbered affiliations see end of article.

**Correspondence to**
Dr Zhong-Chun Liu;
zcliu6@whu.edu.cn

## ABSTRACT

**Objectives** To assess the magnitude of mental health outcomes and associated factors among psychiatric professionals in mental health services during COVID-19 in China.

**Design, setting and participants** This cross-sectional, survey-based, region-stratified study collected demographic data and mental health measurements from psychiatric professionals in 34 hospitals between 29 January and 7 February 2020, in China. Hospitals equipped with fever clinics or deployed on wards for patients with COVID-19 were eligible.

**Primary outcome and measures** The severity of symptoms of depression, anxiety, insomnia and distress were assessed by the Chinese versions of 9-item Patient Health Questionnaire, 7-item Generalised Anxiety Disorder, 7-item Insomnia Severity Index and 22-item Impact of Event Scale-Revised, respectively. Multivariable logistic regression and structural equation modelling was performed to identify factors associated with mental health outcomes.

**Results** A total of 610 psychiatric professionals were included. 29.8% were employed in Wuhan, and 22.5% were frontline workers. A considerable proportion of participants reported symptoms of depression (461 (75.6%)), anxiety (282 (46.2%)), insomnia (336 (55.1%)) and mental stress (481 (78.9%)). Psychiatric symptoms were associated with worrying about infection (eg, OR 2.36 (95% CI 1.27 to 4.39) for anxiety), risks of exposure to COVID-19 (eg, having inadequate personal protection equipment, OR 2.43 (1.32 to 4.47) for depression) and self-perceived physical health (eg, OR 3.22 (2.24 to 4.64) for mental stress). Information sources of COVID-19 were also found to be both positively (eg, information from relatives, OR 2.16 (1.46 to 3.21) for mental stress) and negatively (eg, information from TV, OR 0.52 (0.35 to 0.77) for mental stress) associated with mental stress. There is preliminary evidence that mental health might benefit from greater availability of mental healthcare services. The structural equation model analysis indicated that worrying about infection may be the primary mediator via which risk of exposure to COVID-19 pandemic affects the mental health of psychiatric professionals.

## Strengths and limitations of this study

► Provides a timely evaluation of the acute psychological impact of the COVID-19 pandemic on the mental health of psychiatric professionals in China.

► The large scale nature of the study means it is well powered to detect meaningful effects and allow for the control of potentially confounding variables.

► A sampling bias exists because of the voluntary response sampling method.

► The cross-sectional, observational nature of the study means it is not possible to conclusively determine whether observed associations are causal in nature.

**Conclusions** The current findings demonstrate several pathways via which the COVID-19 pandemic may have negatively affected the mental health of psychiatric professionals in China.

## INTRODUCTION

The 2019 epidemic of COVID-19 that initiated in Wuhan (Hubei Province, China),[1] is now recognised by WHO as a global public health threat.[2] In addition to the resulting increased burden on physical healthcare services, there is mounting recognition of the impact that the pandemic may have on public mental health and therefore mental health services. An effectively functioning mental health service requires sufficient levels of staffing, and staff that are in good physical and mental health themselves.

The mental health of healthcare workers in general has been affected during the immediate wake of the COVID-19 epidemic.[3 4] Psychological distress among medical staff has been noted in the form of increased symptoms of anxiety, depression, insomnia and

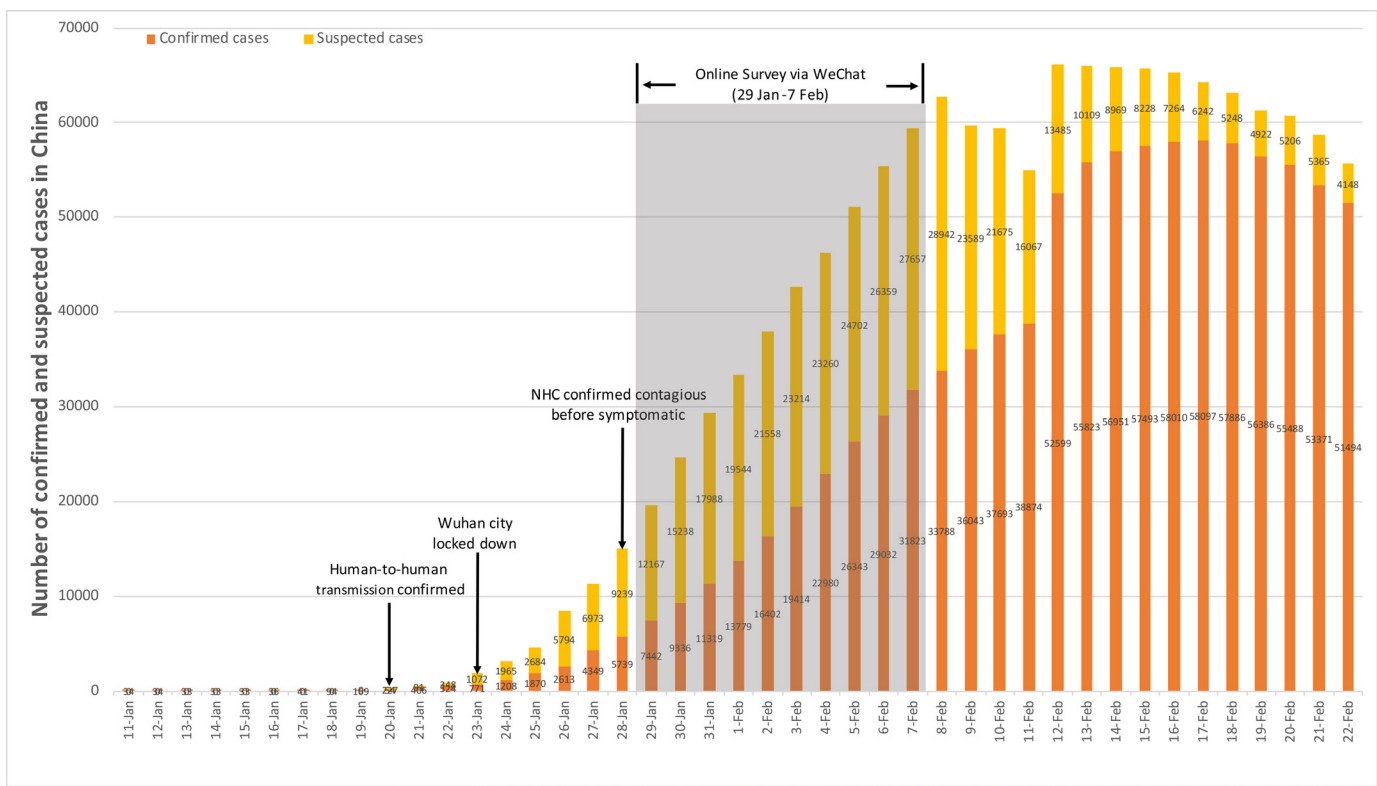

**Figure 1** Timeline of COVID-19 and survey data. Wuhan city was locked down since 23 January 2020, and the online survey was conducted 1 week after that. The cumulative number seemed to have entered the rapid rise phase. NHC, National Health Commission.

post-traumatic stress.[5 6] In addition, workers may also experience isolation from their families or community because of infection transmission risk and stigmatisation.[7]

The specific impact of the pandemic on the mental health of psychiatric professionals is less well characterised. In the current study, a multicentre survey was performed to assess psychological distress in psychiatric professionals in China 1 week after Wuhan city entered lockdown. A brief questionnaire was designed and distributed to psychiatric professionals of several hospitals through the 'WeChat' group chat application, including staff of both psychiatric hospitals and psychiatric staff working in general hospitals and fever clinics. We used the responses from this questionnaire to investigate which factors were associated with an increased risk of mental health symptoms among psychiatric professionals, and whether there was any evidence of potentially protective factors.

## METHODS
### Online survey

A WeChat-based online survey was completed by psychiatric professionals between 29 January and 7 February 2020, the second week after Wuhan city locked down. During the survey period, the cumulative number of confirmed and suspected COVID-19 cases was climbing exponentially (figure 1).

The online survey was conducted with the WeChat-based survey programme 'Questionnaire Star' based on a voluntary response sampling method. The target population was psychiatric professionals working in hospitals that established fever clinics or wards for patients with COVID-19. Each WeChat account of a participant was authenticated by their ID card in China. An online advert with a Quick Response code directing to the online survey was distributed in psychiatric professionals' WeChat working groups of several provinces. Completing the questionnaire outside of working hours was advised. Prior to providing informed consent, potential participants were provided with information regarding the research project including its primary goals. After reading this information, potential participants were provided with the option to either decline or consent to take part in the study. Only those who consented were taken to the questionnaire page. Respondents could terminate the survey at any time or during any question if they wished to.

### Patient and public involvement

No patients were involved in this study.

### Questionnaire

The questionnaire consists of seven sections: basic demographic data, mental health assessment, risks of exposure to COVID-19, source of information about COVID-19, mental healthcare services (MHS) accessed, worry about infection and self-perceived physical health before the

COVID-19 outbreak. The questionnaire required 15 min to complete.

## Demographic data

Basic demographic data include gender, age, marital status (unmarried/other, married), educational level (high school or lower, undergraduate, postgraduate or higher), location, job type (doctor or nurse), job title (junior, intermediate or senior) and hospital type (tertiary, secondary or other).

## Mental health assessment

We used four well-validated scales to assess the mental health status of psychiatric professionals. The 9-item Patient Health Questionnaire (PHQ-9), the 7-item Generalised Anxiety Disorder (GAD-7), the 7-item Insomnia Severity Index (ISI) and the 22-item Impact of Event Scale-Revised (IES-R) were used to evaluate depression, anxiety, insomnia and stress, respectively. The PHQ-9 is a self-report measure used to assess the severity of depression and has good reliability and validity in Chinese populations (Cronbach's α=0.89).[8] PHQ-9 total scores range from 0 to 27, and a score of 10 or greater is defined as a high probability of meeting diagnostic criteria for depression.[9] The GAD-7 is a self-rated scale to evaluate the severity of anxiety and has good reliability and validity in Chinese population (Cronbach's α=0.89),[10] the total score of GAD-7 ranges from 0 to 21 and a score of >10 indicates a high probability of meeting diagnostic criteria for anxiety.[11] The ISI is a measure of insomnia severity that has been shown to be valid and reliable, and a total score of >10 is defined as a high probability of meeting diagnostic criteria for insomnia.[12] Post-traumatic symptoms were accessed using the IES-R, which also has extensive reliability and validity (Cronbach's α for 3 subscales were 0.89 (intrusion), 0.85 (avoidance) and 0.83 (hyperarousal), respectively),[13] and a score of >34 is identified as significant mental stress.[14]

## Risk of exposure to COVID-19

Exposure to COVID-19 was determined by the following questions asked to psychiatric professionals:

1. Have you been diagnosed with COVID-19?
2. Do you manage patients diagnosed with COVID-19?
3. Has your family been diagnosed with COVID-19?
4. Have your friends been diagnosed?
5. Have your neighbours (people living in the same community who may or may not know each other) been diagnosed?
6. Does anyone living with you have suspected symptoms?

In terms of personal protection, participants were asked whether they had enough personal protection equipment (PPE) and enough training for personal protection. The possible responses to each question were yes or no.

## Accessed mental health services

The following question was used to determine which psychological services the subject had accessed: "Have you ever received the following services: psychological support materials (leaflets, brochures and books provided by mental health workers and distributed to staff in the hospital), psychological resources available through media (psychological assistance methods and techniques provided by psychologists through social media) or online counselling or psychotherapy (including individual or group counselling or structural psychotherapy)?"

## Source of information about COVID-19

The following question was used to determine from which source subjects had accessed information regarding COVID-19: "Which way do you used to get information about COVID-19: relatives (information through direct communication with family, neighbours, colleagues, friends or strangers), social media (information from unofficial accounts in WeChat Moments, Weibo, TikTok and other digital platform), TV (information from national and local official television channels) and other (other information sources of COVID-19) and daily time (hours) spending on information of COVID-19". Then, participants were asked how helpful the information they received were. Answer options included considerably helpful, medium helpful, slightly helpful or not helpful at all.

## Self-perceived physical health

Physical health status was determined by asking participants to compare their health status to normal people before the outbreak of COVID-19: "How do you perceive your health status compared with normal people before the outbreak?" Answer options included strong, normal or weak.

## Worrying about infection

Worry about infection was determined by asking participants if they worry about their vulnerability to infection of COVID-19 in workplace or at home. Possible responses were yes or no.

## Statistical analysis

All statistical analyses were carried out in the statistical programming language R (V.3.6.0).[15]

A bivariate analysis was performed first by using a $\chi^2$ test to determine the effect of six components (demographic data, risks of exposure to COVID-19, source of information about COVID-19, MHS accessed, worry about infection and self-perceived physical health) on prevalence of four mental health symptoms (depression, anxiety, insomnia and mental stress, assessed by using PHQ-9, GAD-7, ISI and IES-R separately) among psychiatric professionals.

Stepwise logistic regression analysis according to Akaike information criterion was then performed to examine the effects of the six components on the four mental health assessment (PHQ-9, GAD-7, ISI and IES-R scores) separately. The significance of association between the potential risk factors and outcomes were checked by analysis of variance (ANOVA) and prediction accuracy of models

were counted by area under the receiver operating characteristic curve.

Finally, a structural equation model (SEM) was constructed via R package lavaan V.0.6–7[16] to explore the relationship between mental health assessment (PHQ-9. GAD-7, ISI and IES-R) and the six components, namely demographic data, risks of exposure to COVID-19, source of information about COVID-19, MHS accessed, worry about infection and self-perceived physical health. The estimation method used weighted least squares with mean and variance adjustment test statistics.[17] We used a Monte Carlo method with 1000 guided resamplings to construct a CI for the estimation effect.[18] In SEM, several criteria, such as ratio of value of $\chi^2$ test of model fit and df <3, root mean square error of approximation (RMSEA) values <0.08 and goodness-to-fit index (GFI) values >0.90, indicate acceptable models .[19]

P values <0.05 indicated that a difference was statistically significant.

## RESULTS

Initially, 834 completed questionnaires were received, with a response rate of 78.5%. The duration (seconds) to complete the questionnaire was recorded by Questionnaire star in WeChat. We excluded 119 questionnaires with a complete duration of <5 min (300 s) or >20 min (1200 s), to ensure that all questions had been well understood and completed consecutively by respondents. We further excluded 105 questionnaires from provinces that had recruited fewer than 20 participants. In total, 224 questionnaires were excluded, leaving 610 questionnaires for analysis. Respondents completing the included questionnaires and excluded questionnaires were comparable in age and sex.

### Demographic characteristics

In total, 610 psychiatric professionals, including 164 (26.9%) doctors and 446 (73.1%) nurses were included in the analysis. It was found that 29.8% of these participants were employed in Wuhan and 53.3% were employed in Hubei provinces other than Wuhan; 22.5% were frontline mental health staff and 53.3% of participants were working in a tertiary hospital. The participants tended to be female (76.1%), aged 26–40 years (63.1%), married (68.2%), with an educational level of undergraduate (91.3%) and have a junior job title (62.1%), as shown in table 1.

### Mental health assessment

According to standard questionnaire cut-offs of the PHQ-9, GAD-7, ISI and IES-R scores, 461 (75.6%) of individuals met criteria for depression, 282 (46.2%) for anxiety, 336 (55.1%) for insomnia and 481 (78.9%) for high levels of mental stress.

### Assessed mental health services

As shown in table 1, 36.3% of participants had received psychological support materials, 50.4% had obtained psychological resources available through media

**Table 1** Demographic characteristics and accessed mental health services

| Variables | | Number | Percentage (%) |
|---|---|---|---|
| Total | | 610 | 100 |
| Gender | Female | 464 | 76.1 |
| | Male | 146 | 23.9 |
| Age (years) | 18–25 | 92 | 15.1 |
| | 26–30 | 198 | 32.5 |
| | 31–40 | 187 | 30.7 |
| | >40 | 133 | 21.8 |
| Marital status | Unmarried/Other | 194 | 31.8 |
| | Married | 416 | 68.2 |
| Education level | High school or less | 18 | 3.0 |
| | Undergraduate | 557 | 91.3 |
| | Graduate or higher | 35 | 5.7 |
| Location | Wuhan | 182 | 29.8 |
| | Hubei except Wuhan | 325 | 53.3 |
| | Guangdong | 50 | 8.2 |
| | Heibei | 25 | 4.1 |
| | Yunan | 28 | 4.6 |
| Job type | Doctor | 164 | 26.9 |
| | Nurse | 446 | 73.1 |
| Job title | Junior | 379 | 62.1 |
| | Intermediate | 158 | 25.9 |
| | Senior | 73 | 12.0 |
| Hospital type | Tertiary | 325 | 53.3 |
| | Secondary or other | 285 | 46.7 |
| Working position | Frontline | 137 | 22.5 |
| | Secondary line | 473 | 77.5 |
| Assessed mental health services | | | |
| Psychological support materials | Yes | 287 | 47.0 |
| | No | 323 | 53.0 |
| MHS available through media | Yes | 358 | 58.7 |
| | No | 252 | 41.3 |
| Counselling or psychotherapy | Yes | 82 | 13.4 |
| | No | 528 | 86.6 |

MHS, mental healthcare services.

and 17.5% had participated in online psychological counselling/psychotherapy.

### Risk factors and protective factors of psychological impact of COVID-19 on psychiatric professionals
#### Bivariate analysis

We found that psychiatric professionals who took part in frontline work possessed higher prevalence of all four mental health symptoms (89.9% vs 71.5%, p<0.001 for depression; 66.4% vs 40.4%, p<0.001 for anxiety; 67.9% vs 51.4%, p<0.001 for insomnia; 69.3% vs 52.0%, p<0.001

for mental stress) compared with those who worked in second-line services. Treating and diagnosing infected patient (95% vs 72.6%, p<0.001 for depression; 75% vs 41.9%, p<0.001 for anxiety; 75% vs 52.1%, p<0.001 for insomnia; 82.5% vs 51.9%, p<0.001 for mental stress), working with infected colleagues (87.9% vs 71.9%, p<0.001 for depression; 67.9% vs 39.8%, p<0.001 for anxiety; 66% vs 52%, p<0.001 for insomnia; 72.1% vs 51.1%, p<0.001 for mental stress), employed in Wuhan (Wuhan vs Hubei except Wuhan vs Other provinces except Hubei: 89% vs 72.6% vs 61.2%, p<0.001 for depression; 65.4% vs 39.1% vs 35%, p<0.001 for anxiety; 70% vs 49% vs 48.5%, p<0.001 for insomnia; 72.5% vs 49.5% vs 46.6, p<0.001 for mental stress), worse self-perceived physical health (strong vs normal vs weak: 60.7% vs 91.5% vs 100%, p<0.001 for depression; 26.4% vs 67.1% vs 88.9%, p<0.001 for anxiety; 34.6% vs 76.7% vs 100%, p<0.001 for insomnia; 39.6% vs 73.1% vs 88.9, p<0.001 for mental stress), worrying about infection (83.5% vs 46.2%, p<0.001 for depression; 54.2% vs 16.9%, p<0.001 for anxiety; 60.6% vs 34.6%, p<0.001 for insomnia; 63.5% vs 27.7%, p<0.001 for mental stress) and obtaining information about COVID-19 from relatives (79.4% vs 71.1%, p=0.02 for depression; 50.9% vs 40.8%, p=0.02 for anxiety; 59% vs 50%, p=0.03 for insomnia; 64.4% vs 46.1%, p<0.001 for mental stress) were also significantly associated with higher prevalence of all four mental health symptoms. Self-infection (100% vs 45.8%, p=0.049) and living with infected neighbour (65.4% vs 39.1, p=0.002) were significantly associated with higher prevalence of anxiety.

Access to mental health services was associated with a lower prevalence of mental health symptoms, both access to psychological support materials (70.7% vs 79.9%, p=0.01 for depression; 38.7% vs 52.9%, p<0.001 for anxiety; 49% vs 61%, p=0.004 for insomnia; 51.6% vs 59.8%, p=0.05 for mental stress) and mental health services accessed through media (71.5% vs 81.3%, p=0.007 for depression; 40.8% vs 54%, p=0.002 for anxiety). Obtaining information about COVID-19 from TV (70.3% vs 81.9%, p=0.001 for depression; 41.7% vs 51.6%, p=0.02 for anxiety; 51% vs 60%, p=0.02 for insomnia; 50.5% vs 62.5%, p=0.004 for mental stress) was also significantly associated with lower prevalence of mental health symptoms, as shown in table 2.

## Logistic regression
To explore potential risk and protective factors, stepwise logistic regression analyses on PHQ-9, GAD-7, ISI and IES-R scores separately was performed. After controlling for potential confounding variables, checking the significance of effect of each variable using ANOVA, we identified risk and protective factors as follows (table 3).

### 9-Item Patient Health Questionnaire
Treating and diagnosing infected patient (OR 2.77; 95% CI 1.05 to 7.27; p=0.039), worrying about infection (OR 0.33; 95% CI 0.21 to 0.52; p<0.001), having inadequate PPE (OR 2.43; 95% CI 1.32 to 4.47; p=0.005) and

worse self-perceived physical health (OR 2.76; 95% CI 1.73 to 4.41; p<0.001) were significantly associated with higher PHQ-9 score among psychiatric professionals. Obtaining information about COVID-19 from relatives (OR 1.64; 95% CI 1.03 to 2.61; p=0.036) was also significantly associated with increased PHQ-9 score, while obtaining information from TV (OR 0.57; 95% CI 0.36 to 0.91; p=0.02) was significantly associated with lower PHQ-9 score.

### 7-Item Generalised Anxiety Disorder
Doctors were found to have higher GAD-7 scores (OR 0.55; 95% CI 0.34 to 0.91; p=0.019) compared with nurses. Self-infection seems a primary risk factor to anxiety (OR 2530278.72; 95% CI 0 to infinity; p=0.045), and all self-infection psychiatric professionals met criteria of severe anxiety. Treating and diagnosing infected patient (OR 2.36; 95% CI 1.27 to 4.39; p=0.007), working with infected colleague (OR 2.17; 95% CI 1.31 to 3.61; p=0.003), living with infected neighbour (OR 2.13; 95% CI 1.14 to 3.96; p=0.018), having inadequate PPE (OR 1.7; 95% CI 1.09 to 2.65; p=0.02), worrying about infection (OR 0.25; 95% CI 0.14 to 0.44; p<0.001) and obtaining information from other methods (OR 2.36; 95% CI 1.29 to 4.32; p=0.005) were associated with increased GAD-7 score. Mental health services accessed through psychological support materials (OR 0.66; 95% CI 0.43 to 0.99; p<0.001) and those who thought information about COVID-19 was helpful (OR 1.17; 95% CI 1.01 to 1.36; p=0.35) showed lower GAD-7 score.

### 7-Item Insomnia Severity Index
Treating and diagnosing infected patients (OR 1.91; 95% CI 1.06 to 3.46; p=0.032), worse physical health status (OR 5.68; 95% CI 3.92 to 8.25; p<0.001) and those worrying about infection (OR 0.58; 95% CI 0.36 to 0.92; p=0.02) were significantly associated with higher ISI score.

### 22-Item Impact of Event Scale-Revised
Treating and diagnosing infected patient (OR 2.97; 95% CI 1.54 to 5.73; p=0.001), worse self-perceived physical health (OR 3.22; 95% CI 2.24 to 4.64; p<0.001) and worrying about infection (OR 0.35; 95% CI 0.22 to 0.56; p<0.001) were significantly associated with higher IES-R score among psychiatric professionals. Psychiatric professionals obtaining information from relatives showed significantly higher IES-R score (OR 2.16; 95% CI 1.46 to 3.21; p<0.001), while those who obtained information from TV (OR 0.52; 95% CI 0.35 to 0.77; p=0.001), and those who thought information about COVID-19 was helpful (OR 0.52; 95% CI 0.35 to 0.77; p=0.001) showed lower IES-R score.

### Structural equation modelling
Finally, we established an SEM of the association between the seven areas described in methods. The $\chi^2$ test of model fit yielded a value of 729.6, with df=253, p=0.000, $\chi^2$/df=2.88, RMSEA=0.056, GFI=0.909 and Tucker-Lewis index (TLI)=0.973, indicating a good fit. The results showed that gender, the risk factors of exposure, source

**Table 2** Prevalence of acute psychological effects among psychiatric professionals

| Variables | Total | PHQ-9 | | | GAD-7 | | | ISI | | | IES-R | | |
|---|---|---|---|---|---|---|---|---|---|---|---|---|---|
| | | No | Yes | P value | No | Yes | P value | No | Yes | P value | No | Yes | P value |
| Total | 610 | | | | | | | | | | | | |
| Gender | | | | 0.002 | | | 0.09 | | | 0.26 | | | 0.08 |
| Male | 146 | 50 (34.2) | 96 (65.8) | | 88 (60.3) | 58 (39.7) | | 72 (49.3) | 74 (50.7) | | 74 (50.7) | 72 (49.3) | |
| Female | 464 | 99 (21.3) | 365 (78.7) | | 240 (51.7) | 224 (48.3) | | 202 (43.5) | 262 (56.5) | | 195 (42.0) | 269 (58.0) | |
| Physical health | | | | <0.001 | | | <0.001 | | | <0.001 | | | <0.001 |
| Strong | 318 | 125 (39.3) | 193 (60.7) | | 234 (73.6) | 84 (26.4) | | 208 (65.4) | 110 (34.6) | | 192 (60.4) | 126 (39.6) | |
| Normal | 283 | 24 (8.5) | 259 (91.5) | | 93 (32.9) | 190 (67.1) | | 66 (23.3) | 217 (76.7) | | 76 (26.9) | 207 (73.1) | |
| Weak | 9 | 0 (0) | 9 (100) | | 1 (11.1) | 8 (88.9) | | 0 (0) | 9 (100) | | 1 (11.1) | 8 (88.9) | |
| Worrying about infection | | | | <0.001 | | | <0.001 | | | <0.001 | | | <0.001 |
| Yes | 480 | 79 (16.5) | 401 (83.5) | | 220 (45.8) | 260 (54.2) | | 189 (39.4) | 291 (60.6) | | 175 (36.5) | 305 (63.5) | |
| No | 130 | 70 (53.8) | 60 (46.2) | | 108 (83.1) | 22 (16.9) | | 85 (65.4) | 45 (34.6) | | 94 (72.3) | 36 (27.7) | |
| Work position | | | | <0.001 | | | <0.001 | | | <0.001 | | | <0.001 |
| Frontline | 137 | 14 (10.2) | 123 (89.8) | | 46 (33.6) | 91 (66.4) | | 44 (32.1) | 93 (67.9) | | 42 (30.7) | 95 (69.3) | |
| Secondary line | 473 | 135 (28.5) | 338 (71.5) | | 282 (59.6) | 191 (40.4) | | 230 (48.6) | 243 (51.4) | | 227 (48) | 246 (52) | |
| Self-infection | | | | 0.45 | | | 0.049 | | | 0.12 | | | 0.12 |
| No | 605 | 149 (24.6) | 456 (75.4) | | 328 (54.2) | 277 (45.8) | | 274 (45.3) | 331 (54.7) | | 269 (44.5) | 336 (55.5) | |
| Yes | 5 | 0 (0) | 5 (100) | | 0 (0) | 5 (100) | | 0 (0) | 5 (100) | | 0 (0) | 5 (100) | |
| Patient infection | | | | <0.001 | | | <0.001 | | | <0.001 | | | <0.001 |
| No | 530 | 145 (27.4) | 385 (72.6) | | 308 (58.1) | 222 (41.9) | | 254 (47.9) | 276 (52.1) | | 255 (48.1) | 275 (51.9) | |
| Yes | 80 | 4 (5) | 76 (95) | | 20 (25) | 60 (75) | | 20 (25) | 60 (75) | | 14 (17.5) | 66 (82.5) | |
| Family member infection | | | | 0.28 | | | 0.84 | | | 0.21 | | | 0.65 |
| No | 603 | 149 (24.7) | 454 (75.3) | | 325 (53.9) | 278 (46.1) | | 273 (45.3) | 330 (54.7) | | 267 (44.3) | 336 (55.7) | |
| Yes | 7 | 0 (0) | 7 (100) | | 3 (42.9) | 4 (57.1) | | 1 (14.3) | 6 (85.7) | | 2 (28.6) | 5 (71.4) | |
| Colleague infection | | | | <0.001 | | | <0.001 | | | 0.003 | | | <0.001 |
| No | 470 | 132 (28.1) | 338 (71.9) | | 283 (60.2) | 187 (39.8) | | 227 (48) | 243 (52) | | 230 (48.9) | 240 (51.1) | |
| Yes | 140 | 17 (12.1) | 123 (87.9) | | 45 (32.1) | 95 (67.9) | | 47 (34) | 93 (66) | | 39 (27.9) | 101 (72.1) | |
| Friend infection | | | | 0.44 | | | 0.14 | | | 0.18 | | | 0.22 |
| No | 571 | 142 (24.9) | 429 (75.1) | | 312 (54.6) | 259 (45.4) | | 261 (46) | 310 (54) | | 256 (44.8) | 315 (55.2) | |
| Yes | 39 | 7 (17.9) | 32 (82.1) | | 16 (41) | 23 (59) | | 13 (33) | 26 (67) | | 13 (33) | 26 (67) | |
| Neighbour infection | | | | 0.13 | | | 0.002 | | | 0.61 | | | 0.27 |
| No | 533 | 136 (25.5) | 397 (74.5) | | 300 (56.3) | 233 (43.7) | | 242 (45) | 291 (55) | | 240 (45) | 293 (55) | |
| Yes | 77 | 13 (16.8) | 64 (83.1) | | 28 (36.4) | 49 (63.6) | | 32 (42) | 45 (58) | | 29 (37.7) | 48 (62.3) | |
| Location | | | | <0.001 | | | <0.001 | | | <0.001 | | | <0.001 |

Continued

**Table 2** Continued

| Variables | Total | PHQ-9 | | | GAD-7 | | | ISI | | | IES-R | | |
|---|---|---|---|---|---|---|---|---|---|---|---|---|---|
| | | No | Yes | P value | No | Yes | P value | No | Yes | P value | No | Yes | P value |
| Wuhan | 182 | 20 (11) | 162 (89) | | 63 (34.6) | 119 (65.4) | | 54 (30) | 128 (70) | | 50 (27.5) | 132 (72.5) | |
| Hubei except Wuhan | 325 | 89 (27.4) | 236 (72.6) | | 198 (60.9) | 127 (39.1) | | 167 (51) | 158 (49) | | 164 (50.5) | 161 (49.5) | |
| Other provinces except Hubei | 103 | 40 (38.8) | 63 (61.2) | | 67(65) | 36(35) | | 53 (51.5) | 50 (48.5) | | 55 (53.4) | 48 (46.6) | |
| MHS psychological materials | | | | 0.01 | | | <0.001 | | | 0.004 | | | 0.05 |
| No | 323 | 65 (20.1) | 258 (79.9) | | 152 (47.1) | 171 (52.9) | | 127 (39) | 196 (61) | | 130 (40.2) | 193 (59.8) | |
| Yes | 287 | 84 (29.3) | 203 (70.7) | | 176 (61.3) | 111 (38.7) | | 147 (51) | 140 (49) | | 139 (48.4) | 148 (51.6) | |
| MHS available through media | | | | 0.007 | | | 0.002 | | | 0.11 | | | 0.15 |
| No | 252 | 47 (18.7) | 205 (81.3) | | 116 (46) | 136 (54) | | 103 (41) | 149 (59) | | 102 (40.5) | 150 (59.5) | |
| Yes | 358 | 102 (28.5) | 256 (71.5) | | 212 (59.2) | 146 (40.8) | | 171 (48) | 187 (52) | | 167 (46.6) | 191 (53.4) | |
| Information from relatives | | | | 0.02 | | | 0.02 | | | 0.03 | | | <0.001 |
| No | 284 | 82 (28.9) | 202 (71.1) | | 168 (59.2) | 116 (40.8) | | 141 (50) | 143 (50) | | 153 (53.9) | 131 (46.1) | |
| Yes | 326 | 67 (20.6) | 259 (79.4) | | 160 (49.1) | 166 (50.9) | | 133 (41) | 193 (59) | | 116 (35.6) | 210 (64.4) | |
| Information from TV | | | | 0.001 | | | 0.02 | | | 0.02 | | | 0.004 |
| No | 277 | 50 (18.1) | 227 (81.9) | | 134 (48.4) | 143 (51.6) | | 110 (40) | 167 (60) | | 104 (37.5) | 173 (62.5) | |
| Yes | 333 | 99 (29.7) | 234 (70.3) | | 194 (58.3) | 139 (41.7) | | 164 (49) | 169 (51) | | 165 (49.5) | 168 (50.5) | |

GAD-7, 7-item Generalised Anxiety Disorder; IES-R, 22-item Impact of Event Scale-Revised; ISI, 7-item Insomnia Severity Index; MHS, mental healthcare services; PHQ-9, 9-item Patient Health Questionnaire.

**Table 3** Risk and protective factors of mental health of psychiatric professionals identified by stepwise LR

| Variables | Crude OR (95% CI) | Crude P value | Adjusted OR (95% CI) | P (Wald's test) | P (LR test) |
|---|---|---|---|---|---|
| PHQ-9 | | | | | |
| Gender: male vs female | 0.6 (0.39 to 0.92) | 0.019 | 0.63 (0.38 to 1.03) | 0.063 | 0.066 |
| Physical health (cont. var.) | 3.9 (2.51 to 6.06) | <0.001 | 2.76 (1.73 to 4.41) | <0.001 | <0.001 |
| Patient infection: yes vs no | 4.58 (1.81 to 11.58) | 0.001 | 2.77 (1.05 to 7.27) | 0.039 | 0.022 |
| MHS media: yes vs no | 0.65 (0.43 to 0.98) | 0.039 | 0.69 (0.43 to 1.09) | 0.112 | 0.11 |
| MHS psychotherapy: no vs yes | 3.84 (0.5 to 29.46) | 0.196 | 4.05 (0.49 to 33.61) | 0.195 | 0.127 |
| Having enough PP: no vs yes | 3.15 (1.78 to 5.59) | <0.001 | 2.43 (1.32 to 4.47) | 0.005 | 0.003 |
| Worrying about infection: no vs yes | 0.21 (0.13 to 0.32) | <0.001 | 0.33 (0.21 to 0.52) | <0.001 | <0.001 |
| Information from relatives: yes vs no | 1.81 (1.22 to 2.68) | 0.003 | 1.64 (1.03 to 2.61) | 0.036 | 0.035 |
| Information from TV: yes vs no | 0.63 (0.42 to 0.93) | 0.022 | 0.57 (0.36 to 0.91) | 0.02 | 0.018 |
| GAD-7 | | | | | |
| Gender: male vs female | 0.71 (0.48 to 1.03) | 0.071 | 0.62 (0.37 to 1.03) | 0.066 | 0.064 |
| Physical health (continuous variable) | 5.62 (3.99 to 7.92) | <0.001 | 4.96 (3.37 to 7.29) | <0.001 | <0.001 |
| Job type: doctor vs nurse | 0.84 (0.59 to 1.2) | 0.343 | 0.55 (0.34 to 0.91) | 0.019 | 0.018 |
| Self-infection: yes vs no | 2508169.79 (0 to infinity) | 0.97 | 2530278.72 (0 to infinity) | 0.979 | 0.045 |
| Patient infection: yes vs no | 4.16 (2.44 to 7.1) | <0.001 | 2.36 (1.27 to 4.39) | 0.007 | 0.006 |
| Colleague infection: yes vs no | 3.19 (2.14 to 4.77) | <0.001 | 2.17 (1.31 to 3.61) | 0.003 | 0.002 |
| Friend infection: yes vs no | 1.73 (0.9 to 3.35) | 0.102 | 0.48 (0.2 to 1.11) | 0.086 | 0.087 |
| Neighbour infection: yes vs no | 2.25 (1.37 to 3.7) | 0.001 | 2.13 (1.14 to 3.96) | 0.018 | 0.016 |
| MHS psychological support material: yes vs no | 0.56 (0.41 to 0.77) | <0.001 | 0.66 (0.43 to 0.99) | 0.045 | 0.045 |
| Having enough PP: no vs yes | 2.22 (1.53 to 3.23) | <0.001 | 1.7 (1.09 to 2.65) | 0.02 | 0.02 |
| Worrying about infection: no vs yes | 0.17 (0.11 to 0.28) | <0.001 | 0.25 (0.14 to 0.44) | <0.001 | <0.001 |
| Information from TV: yes vs no | 0.67 (0.49 to 0.93) | 0.015 | 0.73 (0.49 to 1.1) | 0.131 | 0.131 |
| Information from other methods: yes vs no | 1.46 (0.91 to 2.35) | 0.119 | 2.36 (1.29 to 4.32) | 0.005 | 0.005 |
| Thinking Information helpful? (cont. var.) | 1.33 (1.18 to 1.49) | <0.001 | 1.17 (1.01 to 1.36) | 0.035 | 0.035 |
| ISI | | | | | |
| Gender: male vs female | 0.79 (0.55 to 1.15) | 0.221 | 0.97 (0.62 to 1.5) | 0.88 | 0.88 |
| Education (cont. var.) | 1.44 (0.82 to 2.5) | 0.201 | 1.76 (0.95 to 3.27) | 0.073 | 0.07 |
| Physical health (cont. var.) | 6.35 (4.46 to 9.06) | <0.001 | 5.68 (3.92 to 8.25) | <0.001 | <0.001 |
| Patient infection: yes vs no | 2.76 (1.62 to 4.71) | <0.001 | 1.91 (1.06 to 3.46) | 0.032 | 0.027 |
| MHS psychological support material: yes vs no | 0.62 (0.45 to 0.85) | 0.003 | 0.74 (0.51 to 1.07) | 0.111 | 0.111 |
| MHS psychotherapy: no vs yes | 1.65 (0.56 to 4.89) | 0.366 | 2.63 (0.8 to 8.63) | 0.112 | 0.103 |
| Having enough training: no vs yes | 1.95 (1.37 to 2.79) | <0.001 | 1.45 (0.95 to 2.21) | 0.088 | 0.087 |
| Having enough PP: no vs yes | 1.99 (1.36 to 2.91) | <0.001 | 1.4 (0.9 to 2.2) | 0.135 | 0.134 |
| Worrying about infection: no vs yes | 0.34 (0.23 to 0.52) | <0.001 | 0.58 (0.36 to 0.92) | 0.020 | 0.019 |
| Information from social media: yes vs no | 2.08 (0.75 to 5.81) | 0.16 | 2.76 (0.81 to 9.4) | 0.103 | 0.093 |
| Information from TV: yes vs no | 0.68 (0.49 to 0.94) | 0.019 | 0.7 (0.49 to 1.02) | 0.067 | 0.066 |
| IES-R | | | | | |
| Education (cont. var.) | 1.54 (0.88 to 2.7) | 0.129 | 1.58 (0.84 to 2.97) | 0.154 | 0.15 |
| Gender: male vs female | 0.71 (0.49 to 1.02) | 0.067 | 0.81 (0.52 to 1.25) | 0.34 | 0.341 |
| Physical health (cont. var.) | 4.1 (2.93 to 5.74) | <0.001 | 3.22 (2.24 to 4.64) | <0.001 | <0.001 |
| Patient infection: yes vs no | 4.37 (2.4 to 7.98) | <0.001 | 2.97 (1.54 to 5.73) | 0.001 | <0.001 |
| Having enough training: no vs yes | 1.92 (1.35 to 2.75) | <0.001 | 1.42 (0.95 to 2.14) | 0.089 | 0.089 |
| Worrying about infection: no vs yes | 0.22 (0.14 to 0.34) | <0.001 | 0.35 (0.22 to 0.56) | <0.001 | <0.001 |
| Information from relatives: yes vs no | 2.11 (1.53 to 2.93) | <0.001 | 2.16 (1.46 to 3.21) | <0.001 | <0.001 |
| Information from TV: yes vs no | 0.61 (0.44 to 0.85) | 0.003 | 0.52 (0.35 to 0.77) | 0.001 | <0.001 |
| Think information helpful? (cont. var.) | 1.24 (1.1 to 1.39) | <0.001 | 1.15 (1 to 1.31) | 0.045 | 0.044 |

GAD-7, 7-Item Generalised Anxiety Disorder; IES-R, 22-item Impact of Event Scale-Revised; ISI, 7-item Insomnia Severity Index; LR, logistic regression; MHS, mental healthcare services; PHQ-9, 9-Item Patient Health Questionnaire; PP, personal protection.

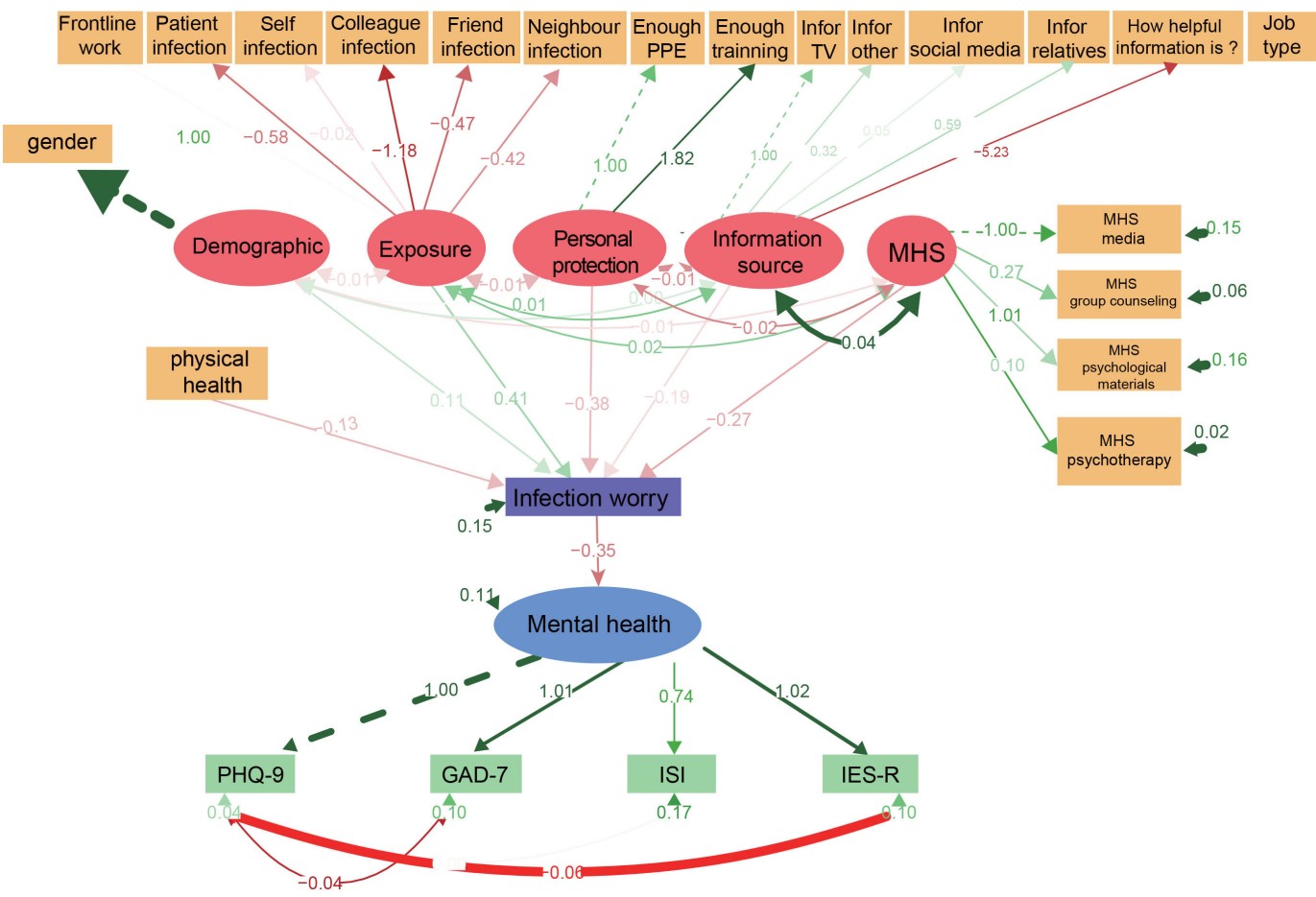

**Figure 2** Direct and indirect effects in SEM. The solid line represents a significant relationship between the two, while the dotted line represents the relationship is not significant. GAD-7, 7-item Generalised Anxiety Disorder; IES-R, 22-item Impact of Event Scale-Revised; ISI, Insomnia Severity Index; MHS, mental healthcare services; PHQ-9, 9-item Patient Health Questionnaire; 7-item.

of information about COVID-19 and self-perceived physical health status affected mental health through worry about infection, indicated that worrying about infection could be the primary mediator that affects the mental health of psychiatric professionals during the COVID-19 outbreak. Mental health services alleviated mental health symptoms through decreasing the worry about infection. The results are shown in figure 2.

## DISCUSSION

The current study investigated the psychological impact of COVID-19 among psychiatric professionals in China during the first months of the pandemic. The overall prevalence of depression, anxiety, insomnia and stress in psychiatric in the second week after Wuhan city locked down were 75.6%, 46.2%, 55.1% and 78.9%, respectively.

In comparison with recent studies of general healthcare workers,[6 20 21] the psychological impact of COVID-19 outbreak seems to be potentially more severe among psychiatric professionals (the overall prevalence of depression, anxiety, insomnia and stress was estimated at 13.9%–46.9%, 16.1%–41.1%, 19.7% and 9.8%–69.1%, respectively). When compared, however, with studies that

have focused on similar regions (ie, Hubei province), the rates appear comparable to other healthcare workers (82.9%–88.7%, 66.9%–80.4% and 87.6%–92.9%, for depression, anxiety and stress, respectively).[20]

We identified several factors that were associated with higher psychiatric symptom burden. An increased risk of exposure to COVID-19, including treating and diagnosing infected patients was significantly associated with higher risk of mental health symptoms among psychiatric professionals. We also found that female healthcare workers had a higher risk of developing depression symptoms but not anxiety, insomnia or mental stress symptoms during the COVID-19 pandemic among psychiatric professionals. A recent meta-analysis study showed that females had a higher risk of developing psychiatric symptoms and reported lower psychological well-being during the COVID-19 pandemic, but the findings of the studies included in the published meta-analysis were not consistent.[22] How much of the association is specific to COVID-19, as opposed to a generally increased risk remains to be established.

This is the first study to explore the psychological effect of different sources of information about COVID-19 on

mental health among psychiatric professionals. At the early stage of the outbreak, the unpredictable nature of the pandemic was accompanied by misinformation, often driven by erroneous news reports from unofficial information sources and the public's misunderstanding of health messages, thus causing worry and increasing the risk of mental health problems in population.[3] Our study suggests that information from social media, direct communication with relatives and other unofficial information sources is associated with an increased risk of mental health symptoms among psychiatric professionals. Conversely, information obtained from official sources such as TV was associated with reduced symptom severity in psychiatric professionals.

To efficiently cope with psychological impact of the COVID-19 outbreak, the Chinese Government has implemented rapid and comprehensive psychological crisis interventions, including publishing psychological support materials, such as leaflets and books, establishing psychological intervention teams on site,[4] an online mental health service through social media platforms and also telephone guidance to help address mental health problems.[23] Our study found that the use of some of these interventions (psychological materials and psychological help from media, and online counselling and psychotherapy) was associated with reduced symptom severity.

Our study also has some limitations. First, the study is cross-sectional and does not record the mental health status before the outbreak, and cannot establish causal associations. There is a potential for a response bias given the voluntary nature of the survey. However, at least as regards occupation and gender, our sample is not substantially different from the overall workforce. Certain factors may limit the generalisability of our findings. Our study was performed in the early days of the outbreak and was concluded by the end of the second week after Wuhan city locked down. During this period, the rapid rise in number of identified and suspected cases and unprecedented scale of the lock down caused public panic and psychological stress, which may have diminished over time.[3]

In summary, the results demonstrate that a strikingly large portion of psychiatric professionals in China suffered from mental health disturbances during the early outbreak of COVID-19. Their mental health was primarily affected by worry about being infected, risks of exposure to COVID-19 and self-perceived physical health. Unreliable information sources of COVID-19 may exacerbate mental health symptoms, and healthcare worker mental health may benefit from greater availability of mental healthcare from public and personalised mental health services.

**Author affiliations**
[1]Department of Psychiatry, Renmin Hospital of Wuhan University, Wuhan, China
[2]Institute of Psychiatry, Psychology and Neuroscience, King's College London, London, UK
[3]Shanghai Clinical Research Center for Mental Health, Shanghai Key Laboratory of Psychotic Disorders, Shanghai Mental Health Center, Shanghai Jiao Tong University School of Medicine, Shanghai, China
[4]School of Computer Science & Technology, Huazhong University of Science and Technology, Wuhan, China
[5]Department of Psychiatry, Zhejiang University, Hangzhou, China

**Contributors** XG, RM and Z-CL had full access to all the data in the study and takes responsibility for the integrity of the data and the accuracy of the data analysis. Design and conduct of the study: XG, RM and Z-CL. Collection, management, analysis and interpretation of the data: XG, RM, TP, AA, JC, SM, JY, YW and SH. Preparation, review or approval of the manuscript: XG, RM, AA, JC, Z-CL and YW. Decision to submit the manuscript for publication: Z-CL and YW.

**Funding** XG's work is funded by the National Key R&D Program of China (no. 2018YFC1314600), and fellowship grants from Chinese Scholarship Council. ZCL's work is funded by the National Key R&D Program of China (no. 2018YFC1314600). RM's research is funded by the Wellcome Trust (no. 200102/Z/15/Z) and a NIHR clinical lectureship.

**Competing interests** None declared.

**Patient and public involvement** Patients and/or the public were involved in the design, or conduct, or reporting, or dissemination plans of this research. Refer to the 'Methods' section for further details.

**Patient consent for publication** Not required.

**Ethics approval** This study was reviewed and approved by the Clinical Research Ethics Committee of Renmin Hospital of Wuhan University (WDRY2020-K004). All subjects provided informed consent electronically prior to registration.

**Provenance and peer review** Not commissioned; externally peer reviewed.

**Data availability statement** All data relevant to the study are included in the article or uploaded as supplementary information.

**ORCID iDs**
Xin Guo http://orcid.org/0000-0002-5482-6612
Zhong-Chun Liu http://orcid.org/0000-0001-5410-0312

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
