## [Reviewer comments · BMJ Open]

ARTICLE DETAILS

TITLE (PROVISIONAL)	Acute psychological impact of Coronavirus Disease 2019 outbreak among psychiatric professionals in China : A multicenter, cross-sectional, web-based study
AUTHORS	Liu, Zhong-Chun; Guo, Xin; McCutcheon, Robert; Pillinger, Toby; Arumham, Atheeshaan; Chen, Jianhua; Ma, Simeng; Yang, Jun; Wang, Ying; Hu, Shaohua; Wang, Gaohua

VERSION 1 – REVIEW

REVIEWER	Herrera-Peco, Iván Universidad Alfonso X el Sabio Facultad de Ciencias de la Salud, Nursery department
REVIEW RETURNED	23-Dec-2020

GENERAL COMMENTS	I have some questions about your manuscript: 1.- I have some concerns about the ethical aspects of your manuscript. It was impossible for me to find any mention about how "possible participants" could read information, including the main goal(s), of your research project. Finally, it would be necessary to include information about how participants give their consent to participate in your study. 2.- When authors explain the "online survey", it would be necessary to explain how they achieve the target population and, thus what kind of sampling method has been used (probably a non-probabilistic one, which must be included as a limitation of this study). 3.- Please, could be possible to Include information about what kind of study are authors developing? in example, It is a quantitative, observational and cross-sectional study? 4.- Results: + The sample included in this study have a 76.1% of male versus 23.9% of females but authors don't include analysis to verify the "possible effect" of sex in ISI and IES-R questionnaires, why? Could you explain? + Authors include information about the "Structural Equation Modelling", where they said: "The results showed that gender, the risk factors of exposure, source of information about COVID-19 and physical health perception affected mental health". But when I read the discussion, I can't find any reference to gender...This meant that I could be interesting data that authors clarify if female, or male, are more susceptible to the risk factors described.
---

	5.- About Limitations:  - Why authors don't mentioned that most of the sample was made up by men? Could authors explain if exist a good invariance across sex in the different questionnaires used in their study? - Please, define as a limitation the sampling methods and their limited capability to extrapolate the conclusions to the population.
REVIEWER	Miniati , M Universita degli Studi di Pisa Dipartimento di Medicina Clinica e Sperimentale, Clinical and Experimental Medicine
REVIEW RETURNED	13-Mar-2021
GENERAL COMMENTS	The manuscript is on an interesting issue and certainly falls within the scope of the Journal. The database is interesting and results are presented clearly. No shortcomings to report.

VERSION 1 – AUTHOR RESPONSE

Reviewer Reports:

Reviewer: 1

Dr. Iván Herrera-Peco, Universidad Alfonso X el Sabio Facultad de Ciencias de la Salud

Comments to the Author:

Dear authors

I have some questions about your manuscript:

1.- I have some concerns about the ethical aspects of your manuscript. It was impossible for me to find any mention about how "possible participants" could read information, including the main goal(s), of your research project.

Finally, it would be necessary to include information about how participants give their consent to participate in your study.

-Thank you for your insightful comments. We have included the information of informed consent in the method.

P8 L4-9

"Prior to providing informed consent potential participants were provided with information regarding the research project including its primary goals. After reading this information potential participants were provided with the option to either decline or consent to take part in the study. Only those who consented were taken to the questionnaire page."

2.- When authors explain the "online survey", it would be necessary to explain how they achieve the target population and, thus what kind of sampling method has been used (probably a non-probabilistic one, which must be included as a limitation of this study).

-Thanks. We have added the information of sampling method in the "online survey".

P7 L17-22 P8 L1

"The online survey was conducted with the WeChat-based survey program 'Questionnaire Star' based on voluntary response sampling method. The target population was psychiatric professionals working in hospitals that established fever clinics or wards for patients with COVID-19. Each WeChat

account of a participant was authenticated by their ID card in China. The online adverts with a Quick Response (QR) code directing to the online survey was distributed in psychiatric professionals' WeChat working groups of several provinces."

3.- Please, could be possible to Include information about what kind of study are authors developing? in example, It is a quantitative, observational and cross-sectional study?

-Thanks. We have revised the title as followed:

P1 L1-2

"Acute psychological impact of Coronavirus Disease 2019 outbreak among psychiatric professionals in China : A multicenter, cross-sectional, web-based study"

4.- Results:

+ The sample included in this study have a 76.1% of male versus 23.9% of females but authors don't include analysis to verify the "possible effect" of sex in ISI and IES-R questionnaires, why? Could you explain?

-Thanks for your insightful comments. The sample included in this study have a 76.1% of female versus 23.9% of males. We have analyzed the possible effect of all potential factors including gender (highlighted) on ISI and IES-R questionnaires, and the bivariate analysis does not demonstrate any significant association between gender and ISI and IES-R (Table 2). We have now also included gender in the step-wised multiple regression analysis (table 3).

P28-29

Table 2. Prevalence of acute psychological effects among psychiatric professionals

P32

Table3. Risk & protective factors of mental health of psychiatric professionals identified by stepwise logistic regression

+ Authors include information about the "Structural Equation Modelling", where they said: "The results showed that gender, the risk factors of exposure, source of information about COVID-19 and physical health perception affected mental health".

But when I read the discussion, I can't find any reference to gender...This meant that I could be interesting data that authors clarify if female, or male, are more susceptible to the risk factors described.

-Thanks. We have added the related discussion on female gender influence on mental health during COVID-19 among psychiatric professionals in the discussion section.

P20 I17-22

"Furthermore, our study found that females had a higher risk of developing depression symptoms but not anxiety, insomnia or mental stress symptoms during the COVID-19 pandemic among psychiatric professionals. A recent meta-analysis study showed that females had a higher risk of developing psychiatric symptoms and reported lower psychological wellbeing during the COVID-19 pandemic, but was not consistent.[22] This may due to various influences such as preconception and fertility issues, pregnancy, postpartum, miscarriage and intimate partner violence.[23] Further studies will be needed to clarify this."

5.- About Limitations:

- Why authors don't mentioned that most of the sample was made up by men? Could authors explain if exist a good invariance across sex in the different questionnaires used in their study?

- Please, define as a limitation the sampling methods and their limited capability to extrapolate the conclusions to the population.

-Thanks. The sample included in this study have a 76.1% of female versus 23.9% of males. We have analyzed the possible effect of all potential factors including gender on ISI and IES-R questionnaires, and the bivariate analysis does not demonstrate any significant association between gender and ISI and IES-R (Table 2). We have now also included gender in the step-wised multiple regression analysis (table 3). Please see our response to Comment on Results section

-Thanks for reminding. We have included this as a limitation in the bullet points and the discussion section as followed:

P5 L7-10

“Possible sampling bias exists because of the voluntary response sampling method “

P22 L10-15

“Second, most respondents (76.1%) in this study were made up by female, this may due to the overall psychiatric workforce was predominantly nurses (73.1%). As by 2019, the ratio of psychiatric doctors to nurses in China was close to 1:2.1, [27] the ratio of doctors to nurses in this study is 1:2.7, while a response bias cannot be ruled out the available data regarding the demographics of the workforce suggests”

Reviewer: 2

Dr. M Miniati, Università degli Studi di Pisa Dipartimento di Medicina Clinica e Sperimentale

Comments to the Author:

The manuscript is on an interesting issue and certainly falls within the scope of the Journal. The database is interesting and results are presented clearly. No shortcomings to report.

-Thank you for your recognitions!

Reviewer: 1

Competing interests of Reviewer: None Declared

Reviewer: 2

Competing interests of Reviewer: None declared